# *Artocarpus lakoocha* Extract Inhibits LPS-Induced Inflammatory Response in RAW 264.7 Macrophage Cells

**DOI:** 10.3390/ijms21041355

**Published:** 2020-02-17

**Authors:** Phateep Hankittichai, Pensiri Buacheen, Pornsiri Pitchakarn, Mingkwan Na Takuathung, Nitwara Wikan, Duncan R. Smith, Saranyapin Potikanond, Wutigri Nimlamool

**Affiliations:** 1Department of Pharmacology, Faculty of Medicine, Chiang Mai University, Chiang Mai 50200, Thailand; phateep.han18@gmail.com (P.H.); mingkwan.n@cmu.ac.th (M.N.T.); saranyapin.p@cmu.ac.th (S.P.); 2Graduate School, Chiang Mai University, Chiang Mai 50200, Thailand; pensiri8@hotmail.com; 3Department of Biochemistry, Faculty of Medicine, Chiang Mai University, Chiang Mai 50200, Thailand; pornsiri.p@cmu.ac.th; 4Research Center of Pharmaceutical Nanotechnology, Chiang Mai University, Chiang Mai 50200, Thailand; 5Institute of Molecular Biosciences, Mahidol University, Salaya, Nakorn Pathom 73170, Thailand; nitwara.wik@mahidol.edu (N.W.); duncan_r_smith@hotmail.com (D.R.S.)

**Keywords:** anti-inflammation, macrophage cells, Artocarpus lakoocha, Akt activation, inflammatory cytokines

## Abstract

*Artocarpus lakoocha* Roxb. (AL) has been known for its high content of stilbenoids, especially oxyresveratrol. AL has been used in Thai traditional medicine for centuries. However, the role of AL in regulating inflammation has not been elucidated. Here we investigated the molecular mechanisms underlying the anti-inflammation of AL ethanolic extract in RAW 264.7 murine macrophage cell line. The HPLC results revealed that this plant was rich in oxyresveratrol, and AL ethanolic extract exhibited anti-inflammatory properties. In particular, AL extract decreased lipopolysaccharide (LPS)-mediated production and secretion of cytokines and chemokine, including IL-6, TNF-α, and MCP-1. Consistently, the extract inhibited the production of nitric oxide (NO) in the supernatants of LPS-stimulated cells. Data from the immunofluorescence study showed that AL extract suppressed nuclear translocation of nuclear factor-kappa B (NF-κB) upon LPS induction. Results from Western blot analysis further confirmed that AL extract strongly prevented the LPS-induced degradation of IκB which is normally required for the activation of NF-κB. The protein expression of iNOS and COX-2 in response to LPS stimulation was significantly decreased with the presence of AL extract. AL extract was found to play an anti-inflammatory role, in part through inhibiting LPS-induced activation of Akt. The extract had negligible impact on the activation of mitogen-activated protein kinase (MAPK) pathways. Specifically, incubation of cells with the extract for only 3 h demonstrated the rapid action of AL extract on inhibiting the phosphorylation of Akt, but not ERK1/2. Longer exposure (24 h) to AL extract was required to mildly reduce the phosphorylation of ERK1/2, p38, and JNK MAPKs. These results indicate that AL extract manipulates its anti-inflammatory effects mainly through blocking the PI3K/Akt and NF-κB signal transduction pathways. Collectively, we believe that AL could be a potential alternative agent for alleviating excessive inflammation in many inflammation-associated diseases.

## 1. Introduction

Inflammation is typically described as a response to stimulation by invading pathogens or endogenous signals, including damaged cells, that results in tissue healing for restoring normal function [1]. Notably, the major immune cells of innate immunity are macrophages, which are elicited for the initiation of inflammatory response affected by overwhelming production of proinflammatory mediators such as nitric oxide (NO), interleukin 6 (IL-6), tumor necrosis factor α (TNF-α), IL-1β, and monocyte chemoattractant protein-1 (MCP-1) [2,3]. Interestingly, much has been discovered about the occurrence of lipopolysaccharide (LPS), the major part of the cell wall of Gram-negative bacteria and the prominent ligand of Toll-like receptor 4 (TLR4), in immoderately initiating the innate immune response resulting in organ failure and even death [4]. 

After ligation with LPS, intracellular signaling is proceeded via the recruitment of the adaptor protein MyD88 and the IL-1R-associated protein kinases (IRAKs), which in turn recruit TNF-receptor associated factor 6 (TRAF6) [5]. Subsequently, the IRAK-TRAF6 complex can propagate the signaling cascades by stimulating four major signaling pathways, including the nuclear factor-kappa B (NF-κB) pathway and three mitogen-activated protein kinase (MAPK) pathways reaching to the activation of extracellular signal-regulated kinases 1/2 (ERK1/2), c-Jun-N-terminal kinases (JNKs), and p38 MAPKs. Besides, ligand binding to TLR4 can activate immune responses that are contributed by the phosphatidylinositol (PI) 3-kinase (PI3K)/Akt signal transduction pathway [2,6]. Likewise, the activation by LPS can stimulate immune cells resulting in the downstream synthesis of pro-inflammatory cytokines, chemokines, and interferons [3]. Thus, controlling the inflammatory cytokines and mediators may have therapeutic potential in the treatment of various inflammatory diseases.

*Artocarpus lakoocha* Roxb. (AL) or Ma-haad has been used in Thai traditional medicine and the major active constituent is oxyresveratrol (*trans*-2,3′,4,5′-tetrahydroxystilbene), a polyphenolic stilbene, which is predominant in the heartwood of the plant [7,8]. Moreover, it has many pharmacological activities, including anti-cancer, anti-tyrosinase, anti-bacterial, anti-viral, anti-oxidant, and anti-inflammatory properties [8,9,10,11,12,13,14,15,16,17,18,19]. Specifically, for anti-inflammatory effects, several studies have revealed that oxyresveratrol may cross the blood–brain barrier in healthy rat, exerts defense of neuroblastoma SH-SY5Y cells against H_2_O_2_-induced toxicity, suppresses the activation of NLRP3 inflammasome, and NF-κB nuclear translocation in primary culture of rat cortical neurons after oxygen-glucose deprivation followed by reperfusion [20,21,22]. However, evidence for whether AL extract suppresses LPS-induced inflammatory signaling in macrophage RAW 267.4 cells has not yet been elucidated. Therefore, we are interested in investigating whether AL possesses anti-inflammatory properties by defining their effects at the molecular level downstream to the LPS-induced inflammatory signaling in a murine macrophage cell line, RAW 264.7. Our study provided accumulated evidence that the ethanolic extract from AL could decrease lipopolysaccharide (LPS)-mediated production of IL-6, TNF-α, monocyte chemotactic protein 1 (MCP-1), and nitric oxide (NO) in RAW 264.7 cells. AL extract was found to effectively attenuate NF-κB activation (particularly by inhibiting the degradation of IκB) and Akt phosphorylation, while moderately reduced the MAPK signal transduction pathway. Since AL extract exhibited the ability to ameliorate the LPS-induced inflammatory response in macrophages, we believe that AL extract could be added to the list of essential natural candidates to be developed as a potential anti-inflammatory agent. 

## 2. Results

### 2.1. Chemical Profile of Oxyresveratrol in Artocarpus Lakoocha Roxb. (AL) Extract 

The major chemical constituents of the ethanolic extract from *Artocarpus lakoocha* were determined by HPLC in comparison to the oxyresveratrol standard compound. The chromatogram of AL extract was identified by comparing the retention time (RT) to that of the standard oxyresveratrol. The HPLC results clearly revealed a major peak of the extract (RT = 27.729) (Figure 1A). This retention time detected in the extract was compatible with that of standard oxyresveratrol (RT = 27.639) (Figure 1B). Additionally, when the mixture of AL extract and oxyresveratrol standard was examined at the same time, the results clearly demonstrated the perfect single peak of oxyresveratrol (RT = 27.753) (Figure 1C). In the comparison with the standard curve, the amount of oxyresveratrol in the extract was found to be 761.8 ± 7.350 mg/g extract, which was then calculated to be approximately 76.2% w/w. These results clearly confirmed that oxyresveratrol with its chemical structure shown in Figure 1D is a major compound, and probably is an active constituent in the extract.

### 2.2. Effects of AL on the Viability of RAW 264.7 Cells

To determine the effects of AL on cell viability, RAW 264.7 cells were treated with the AL extract at different concentrations ranging from 0 to 50 μg/mL for 48 h. The results from 3-(4,5-dimethylthiazol-2-yl)-2,5-diphenyltetrazolium bromide (MTT) assay (Figure 2) showed that the AL extract had some cytotoxic effect on RAW 264.7 cells in a concentration-dependent manner, but the concentration toxic to cells was determined to be more than 25 μg/mL. The IC50 value of AL extract was 31.43 μg/mL. AL extract at 40 μg/mL showed maximum cytotoxic effect, where approximately 90% reduction of RAW 264.7 cell viability was observed. RAW 264.7 cells treated with higher concentrations of AL extract (45 and 50 μg/mL) showed effects on cell viability similar to those treated with AL at 40 μg/mL. However, the AL extract at concentrations below 25 μg/mL did not affect RAW 264.7 cell viability. DMSO (vehicle control), at all concentrations (0% to 0.05%) relevant to the treatment group showed no apparent cytotoxicity to RAW 264.7 cells. To determine anti-inflammatory effects of AL on RAW 264.7 cells, we sought to test the activity of the extract at the non-toxic concentrations which were chosen to be 5, 10, and 20 μg/mL These non-toxic concentrations were used throughout all experiments in this study. 

### 2.3. Effects of AL on Inflammatory Cytokine Production in Lipopolysaccharide (LPS)-Induced RAW 264.7 Cells

RAW 264.7 cells are macrophages which play critical role in the regulation of inflammatory diseases. We investigated whether AL has the ability to regulate the inflammatory response of macrophages in response to LPS stimulation. We utilized ELISA technique to determine the level of nitric oxide (NO), proinflammatory cytokines (IL-6, and TNF-α), and chemokine (MCP-1) in the supernatants collected from cells treated with various concentrations of AL extract and stimulated with LPS for 6 or 24 h. The results showed that AL extract treatment potently inhibited LPS-induced production and secretion of NO (Figure 3A), IL-6 (Figure 3B), TNF-α (Figure 3C), and MCP-1 (Figure 3D) in RAW264.7 cells. These inhibitory effects of AL extract were observed to be in a concentration-dependent manner. These results suggest that AL, containing high amount of oxyresveratrol, could be able to suppress the production and secretion of inflammatory mediators and cytokines from RAW 264.7 cells stimulated with LPS.

### 2.4. Effects of AL on Suppressing Nuclear Factor-kappa B (NF-κB) Activation in LPS-Induced RAW 246.7 Cells

On the basis of our observation from ELISA that AL could potently suppress NO, IL-6, TNF-α, and MCP-1 levels in the culture supernatants of RAW 246.7 cells stimulated with LPS, it is reasonable to hypothesize that AL extract may suppress the activation of NF-κB in response to LPS stimulation. We, therefore, tested whether AL extract inhibits the LPS-induced translocation of NF-κB from the cytoplasm to the nucleus. We performed immunofluorescence study to monitor the localization status of the p65 subunit of NF-κB. Results showed that RAW 246.7 cells without any treatment retained NF-κB in the cytoplasm of the cells (Figure 4a–c), and the cells responded to LPS stimulation clearly revealed that NF-κB translocated into the nucleus (Figure 4d–f) where it normally functions to activate several different genes responsible for the synthesis of inflammatory enzymes, such as iNOS and inflammatory cytokines, and chemokines including IL-6, TNF-α, and MCP-1. Undoubtedly, data demonstrated that AL at 10 μg/mL could dramatically suppress NF-κB nuclear translocation upon LPS induction (Figure 4g–i). These results facilitate us to identify that AL extract exerts its ability to reduce LPS-induced inflammation, at least in part through the inhibition of NF-κB activation. 

We also performed Western blot analysis to confirm whether AL extract suppresses NF-κB nuclear activation by inhibiting the downregulation of IκB upon LPS induction. As expected, the results clearly demonstrated that treating cells for 3 h with AL extract could significantly inhibit LPS-induced degradation of IκB in a concentration-dependent manner (Figure 5A,B). Furthermore, two major inflammatory enzymes, downstream to the NF-κB signaling, which included inducible nitric oxide synthase (iNOS) and cyclooxygenase-2 (COX-2) were detected to be significantly reduced when AL extract was present in cells stimulated with LPS for 24 h (Figure 5C–E). These data verify that AL extract could regulate LPS-induced production of nitric oxide, cytokines, and chemokine via inhibiting IκB degradation and thus suppressing nuclear translocation of NF-κB. 

### 2.5. AL Inhibits LPS-Induced Mitogen-activated Protein Kinase (MAPK) and Akt Signaling Activation

It is well known that in response to LPS stimulation, TLR4 activation leads to recruitment of the adaptor proteins such as myeloid differentiation protein 88 (MyD88) and TIR-domain-containing adaptor protein inducing IFN (TRIF) followed by activation of extracellular signal-regulated kinase 1/2 (ERK1/2), c-Jun N-terminal kinase (JNK), and p38 MAPK pathways, nuclear factor κ B (NF-κB) pathway, and phosphoinositide 3-kinase (PI3K)/Akt signaling pathway [23], resulting in the production of inflammatory cytokines. Therefore, we sought to identify the possible mechanism of actions of AL extract in reducing the production of inflammatory cytokines. The signal transduction pathways of MAPK and Akt were determined in cells treated with AL extract and stimulated with LPS. As expected, we found that AL extract moderately inhibited LPS-induced ERK, JNK, and p38 MAPKs, since its inhibitory effects were seen only after cells were treated with the extract (in the presence of LPS) for 24 h (Figure 6A–D). 

Surprisingly, pretreatment of cells with AL extract at all concentrations tested for 3 h before LPS stimulation for 30 min was not strong enough to suppress the phosphorylation of ERK 1/2 kinase (Figure 7A,C). However, AL extract strongly inhibited phosphorylation of Akt at Ser473, which represents PI3K/Akt pathway activation following LPS stimulation. This observation was seen in cells pretreated with AL extract for only 3 h before LPS activation, and the inhibition of Akt phosphorylation was observed to be in a concentration-dependent manner (Figure 7A,B).

Moreover, data obtained from observation in individual cells by immunofluorescence study clearly confirmed our discovery that phosphorylation of Ser473 of Akt induced by LPS was dramatically inhibited by the action of AL extract at 10 μg/mL (Figure 8g–i) and at 20 μg/mL (Figure 8j–l).

We further examined the strength of AL extract on inhibiting LPS-evoked phosphorylation of Akt (Ser473) at different time points over the 12 h course of LPS activation. Results obviously showed that AL extract almost completely suppressed LPS-evoked phosphorylation of Akt in RAW 264.7 cells at all time points, whereas the phosphorylation of ERK1/2 (pERK1/2) was not affected (Figure 9A–D). Together, these results indicate that pretreatment with AL extract reduces the phosphorylation of Akt (Ser473) at relatively earlier time points.

## 3. Discussion

Although inflammation is a natural defense system of our body, its prolonged existence can cause severe conditions to the patients. Sepsis is one of the conditions which is a life-threatening syndrome. Overwhelming inflammation induced by microbial components or toxins during infection can stimulate overproduction of inflammatory cytokines causing severe condition including sepsis [24]. The most well-characterized sepsis-inducing factor is lipopolysaccharide (LPS), the cell wall constituent of Gram-negative bacteria, activates Toll-like receptor 4 (a member of pattern recognition receptors) [4]. Stimulation with LPS leads to increased secretion of pro-inflammatory cytokines, including IL-6, TNF-α, and IL-1β in the bloodstream, which are resulted from exacerbation of inflammation in patients with sepsis [25]. Many attempts have been focused on finding new strategies for neutralization of bacterial LPS or inhibition of its recognition by host cell receptors. For instance, anti-microbial peptide [26,27] and lipopolyamine [28] have been proposed to be promising drug candidates that modulate the inflammatory signaling at the receptor level. However, many studies have focused on identifying drug candidates that have the ability to manipulate inflammatory signal transduction pathways at the intracellular level downstream of TLR4. For example, Ephedrine hydrochloride (EH) interferes with the production of pro-inflammatory and anti-inflammatory cytokines in TLR4 or TLR2 signaling [29,30]. Medicinal plants are good sources of potential active anti-inflammatory agents. We previously reported various pharmacological effects of certain plants which have been traditionally used in Thailand to treat different diseases, and many parts of their mechanisms of action were associated with the modulation of inflammatory signal transduction pathways [31,32,33,34,35]. 

Recently, extensive studies have been conducted to discover novel effective agents with anti-inflammatory properties. In this study, we were interested in studying anti-inflammatory activities of the extract from *Artocarpus lakoocha* Roxb (AL), which has been known for its high content of stilbenoids such as pinostilbene, desoxyrhapontigenin, pterostilbene, resveratrol, and oxyresveratrol [36]. This plant is called “Ma-haad” and has been used in Thai traditional medicine. There have been studies related to the pharmacological effects of the extract from this plant. One study has reported about its antimicrobial activities [37]. Another study investigated its potent anti-tyrosinase and in vivo skin whitening activities [38]. However, little is known about other pharmacological properties of this plant. Here we accumulated interesting evidence showing that AL extract inhibited LPS-induced production of inflammatory cytokines and chemokine, including IL-6, TNF-α, and MCP-1 as well as an important inflammatory mediator, nitric oxide (NO), in the RAW 267.4 macrophage cell line in response to LPS stimulation. It is well studied that both IL-6 and TNF-α are the primary mediators of local inflammation and sepsis [39]. In particular, a correlation between serum IL-6 and TNF-α level and multiple organ failure has been reported [40,41]. 

Our finding that AL extract could suppress LPS-induced production of IL-6, TNF-α, and MCP-1 suggests that AL extract may be one of the good natural sources for the therapy of inflammation. As a step towards the development of drugs for anti-inflammation treatment, it is important to understand the mechanism of action of AL extract. Therefore, we further investigated the crucial signal transduction pathways related to TLR4 activation in response to LPS stimulation. Normally, LPS activates TLR4 and causes TLR4 ligation. After TLR4 ligation, MyD88 and TRIF are recruited to phosphorylate IRAKs. The TRAF6 interacts with a pre-assembled kinase complex containing TAK1 and TAB1/2/3, resulting in the activation of nuclear factor-kappa B (NF-κB) and mitogen-activated protein kinases (MAPKs), which are necessary for initiating the expression of pro-inflammatory cytokines and chemokines [42]. Data from our study demonstrated that AL extract treatment clearly suppressed the phosphorylation of MAPKs, including, pERK1/2, p-p38, and p-JNK when cells were incubated with AL extract for 24 h. However, this inhibitory effect of AL extract was not observed when cells were pretreated with AL extract for only 3 h. These results indicate that AL extract can minimally suppress the activation of MAPK signaling. The inhibitory efficiency against the MAPK signal transduction pathway was not rapidly regulated by AL extract, but was likely to be in a delayed manner. In addition to MAPKs, PI3K/Akt signaling also contributes to the signal transduction of TLR4 to regulate the expression of inflammatory and anti-inflammatory cytokines [43]. Surprisingly, in contrast to MAPKs, AL extract exhibited strong and rapid inhibitory activity against LPS-induced Akt activation, since a drastic reduction of Akt phosphorylation (Ser473) was observed in cells pretreated with AL extract for only 3 h. We monitored the inhibitory effects of AL extract on Akt phosphorylation for various time points, ranging from 2 to 360 min, after LPS stimulation and found that AL extract could almost completely suppress Akt phosphorylation at all time points, whereas the phosphorylation status of ERK1/2 was not changed. These results verified the possibility that AL extract may exerts its preferential inhibitory action (within a short period of incubation time) towards Akt rather than MAPKs.

Since MAPK and Akt signaling pathways contribute to the activation of NF-κB, at least in part, through the phosphorylation of IκB (an inhibitor of NF-κB), which results in its the degradation via the ubiquitin-proteasome system and eventually leads to NF-κB liberation and translocation into the nucleus, the observation that AL extract inhibited each of these molecular signal transduction pathways was a reasonable explanation for how AL extract suppresses NF-κB activation. Data from Western blot analysis clearly showed that AL extract strongly inhibited the degradation of IκB in RAW 264.7 cells in a concentration-dependent manner after LPS stimulation. Consistently, our immunofluorescence results verified that AL extract could suppress NF-κB translocation from the cytoplasm into the nucleus of the cells in response to LPS addition. Altogether, these results strongly suggest that the down-regulation of IκB and the inhibition of NF-κB activation by AL extract may account for the reduction of the LPS-induced pro-inflammatory cytokines. Besides important cytokines including IL-6 and TNF-α, NF-κB is also well known to play a critical role in inducing the expression of inflammatory enzymes such as iNOS and COX-2 [44,45,46]. Therefore, we detected the level of protein expression of iNOS and COX-2, and the results were as expected revealing that the expression of these two inflammatory enzymes was tremendously reduced in cells treated with AL extract. The observation that iNOS was down-regulated by the action of AL extract may help explain the reason for the reduction of nitric oxide in the supernatants of the treated cells. Additionally, the ability of AL extract to reduce the LPS-induced activation of ERK1/2, p38, and JNK may also contribute to a decrease in iNOS and COX-2. This statement can be supported by the fact that MAPKs have been reported to be involved in regulating iNOS and COX-2 genes, because inhibiting the activity of MAPKs led to suppression of iNOS and COX-2 gene expression [47].

In conclusion, the recent study provides crucial information that the extract from *Artocarpus lakoocha* Roxb, with the enrichment of oxyresveratrol, possesses an activity to inhibit the LPS-stimulated inflammatory response though the blocking of the Akt, MAPK, and NF-κB signaling pathways in RAW 264.7 macrophages, as illustrated in Figure 10. Our data strongly suggest that *Artocarpus lakoocha* Roxb possesses anti-inflammatory effects and could be a good candidate for development as an anti-inflammatory agent.

## 4. Materials and Methods 

### 4.1. Plant Material and Extraction of Artocarpus Lakoocha (AL) Heartwoods

The heartwoods of AL were bought from Thai Lanna Herbal Industry Company Ltd. (Chiang Mai, Thailand) and then authenticated by the specimen of AL (voucher specimen number 006973) from the herbarium of Faculty of Pharmacy, Chiang Mai University (herbarium code: CMU). All steps of AL crude extraction were performed according to a previously published protocol [16]. Briefly, for extraction, the heartwoods of AL which had been dried at 50 °C in hot air oven for 24 h, were ground to powder and then macerated for 6 h in 95% ethanol three times. After maceration, the AL ethanolic extracts were pooled and concentrated using a rotary evaporator under reduced pressure below 45 °C. AL extract was kept in an air-tight and light-protected container until used. 

For the experiment, 1 g of AL was dissolved in 1 mL of dimethyl sulfoxide (DMSO) to make a 1 g/mL stock solution. The AL stock was pre-diluted in medium prior to each treatment. Each experiment was carried out using three independent batches of AL extract. In addition, the final concentration of DMSO was preserved below 0.5% throughout the experiment

### 4.2. High-Performance Liquid Chromatograph Analysis (HPLC)

The AL extract was determined for the existence of oxyresveratrol content by HPLC using a C18 column (250 × 4.6 mm, 5 μm) (Sigma-Aldrich, Saint Louis, MO, USA). The chromatographic separation was carried out using a linear gradient of mobile phase A (1% acetic acid in water) and mobile phase B (100% acetonitrile) for detection. Ten microliters of 500 μg of AL extract dissolved in 1 mL of MeOH was injected into the column with a flow rate of 0.7 mL/min and detection at 330 nm. Peak area and retention time of the extract sample were evaluated as the comparison with standard curve of standard oxyresveratrol (Sigma-Aldrich, Saint Louis, MO, USA) (catalog number 91211).

### 4.3. Cell Culture

The mouse RAW 264.7 murine macrophages (ATCC^®^ TIB-71 ^TM^) used in this study were obtained from ATCC (ATCC, Manasssas, VA, USA). The cells were cultured in complete medium, which was Dulbecco’s modified Eagle’s medium (DMEM) (Gibco, Thermo Fisher Scientific, Waltham, MA, USA), supplemented with 10% fetal bovine serum (Merck KGaA, Darmstadt, Germany) and antibiotics (100 U/mL penicillin and 100 μg/mL streptomycin) (Gibco, Thermo Fisher Scientific, Waltham, MA, USA), and maintained under a humidified atmosphere of 37 °C, 5% CO_2_. The cells were sub-cultured when they reached 80–90% confluence.

### 4.4. Cell Viability Assay

The effect of AL on cell viability was assayed by 3-(4,5-dimethylthiazol-2-yl)-2,5-diphenyltetrazolium bromide (MTT) to obtain the range of toxic and nontoxic concentrations. The MTT assay was performed according to a previously published protocol [48]. RAW264.7 cells were seeded in 96-well plates at a density of 5 × 10^4^ cells per well for 24 h in complete medium. Cells were then treated with AL extract at various concentrations (1–50 μg/mL) or with vehicle (DMSO at 0.001–0.05%) for 48 h, then cells were exposed to the MTT reagent (0.4 mg/mL in PBS) for 30 min at 37 °C, 5% CO_2_. After aspirating the culture supernatants, 200 μL of DMSO was added to each well. The absorbance at 570 nm was measured using a microplate reader (BioTek Instruments, Winooski, VT, USA). The cell viability assay was performed 3 times, and each assay was undertaken in triplicate (n = 9 in three individual experiments). The half maximal inhibitory concentration (IC_50_) value and the range of non-cytotoxic and cytotoxic concentrations of AL extract to the cells were obtained. Three different non-cytotoxic concentrations (5, 10, and 20 µg/mL) were selected for all experiments. 

### 4.5. Nitric Oxide (NO) Production Assay

RAW 264.7 cells were plated at a density of 3 × 10^5^ cells per well in a 24-well plate for 24 h. Cells were then treated for 3 h with three non-cytotoxic concentrations of AL in the presence of 1 μg/mL of lipopolysaccharide (LPS). The 24 h-conditioned media were collected for NO analysis. As an indicator of NO production, the nitrite concentrations were detected by using Griess’ reagent (Sigma-Aldrich, Saint Louis, MO, USA). Equal volumes of supernatants and Griess’ reagent were mixed for 15 min. Present nitrite ions formed a pink diazo dye by diazonium coupling reaction with *N*-(1-Naphthyl) ethylenediamine. Finally, the absorbance was measured spectrophotometrically using a microplate reader (BioTek Instruments, Winooski, VT, USA) at 540 nm and compared with a standard nitrite curve ranging from 0 to 1000 μM.

### 4.6. Enzyme-Linked Immunosorbent Assay (ELISA)

RAW 264.7 cells were left untreated or treated with different nontoxic concentrations of AL with the presence of 1 μg/mL of LPS for 6 and 24 h. The supernatants of the cells were harvested and kept at –20 °C until use. The concentration of IL-6, TNF-α, and MCP-1 in the culture supernatants was measured by using ELISA MAX^TM^ Deluxe Set (BioLegend, San Diego, CA, USA) according to the manufacturer’s protocol. Briefly, the capture antibody diluted in a coating buffer was added into individual wells of a microtiter plate and incubated at 4 °C overnight. Then, the plate was added with blocking buffer for 1 h, at room temperature (RT). Sample supernatants were added into each well and plates were incubated at RT for 2 h. After washing, a detection antibody solution was added to each well for 1 h. Then the plate was washed four times before adding a diluted avidin-HRP solution and incubating for 30 min. The signal was developed by the addition of freshly mixed TMB substrate solution. A stop solution was added to each well, and the absorbance was read at 450 and 570 nm with a microplate reader (BioTek Instruments, Winooski, VT, USA).

### 4.7. Western Blot Analysis

For investigating the effects of AL on inhibiting LPS-induced expression and phosphorylation of importance proteins in NF-κB, MAPK, and PI3K/Akt signaling pathways. RAW 264.7 cells were treated with AL at three different concentrations for 3 h and stimulated with 1 μg/mL of LPS at appropriate time points before harvesting. After treatment, cells were lysed by adding 1× reducing Laemmli buffer. The cell lysates were collected, heated at 100 °C for 5 min, separated by SDS-PAGE, and transferred onto polyvinylidene difluoride (PVDF) membranes (GE Healthcare Life Science, Marlborough, MA, USA). Membranes were blocked with 5% skim milk in a mixture of tris-buffered saline and Tween 20 (TBS-T) (0.02 M Tris-HCl, pH 7.6, 0.0137 M NaCl, and 0.05% Tween 20) (all reagents from Sigma-Aldrich, Saint Louis, MO, USA) at RT for 1 h. Membranes were then incubated with an appropriate primary antibody (Cell Signaling Technology, Boston, MA, USA) diluted in 5% BSA (Merck KGaA, Darmstadt, Germany) in TBS-T at 4 °C overnight. Primary antibodies included a dilution of a phosphospecific rabbit anti-p44/42 MAPK (Erk1/2) (Thr202/Tyr204) antibody (catalog number 4370), a mouse anti-p44/42 MAPK (Erk1/2) antibody (catalog number 9107), a phosphospecific mouse anti-p38 MAPK (Thr180/Tyr182) antibody (catalog number 9216), a rabbit anti-p38 MAPK (D13E1) antibody (catalog number 8690), a phosphospecific rabbit anti-SAPK/JNK (Thr183/Tyr185) antibody (catalog number 4668), a rabbit anti-NF-κB p65 (D14E12) antibody (catalog number 8242), a mouse anti-IκBα (L35A5) antibody (catalog number 4814), a phosphospecific rabbit anti-Akt (Ser473) antibody (catalog number 4060), or a mouse anti-β-actin antibody (catalog number MA1115) (Boster Biological Technology, Pleasanton, CA, USA). After washing with TBS-T, membranes were incubated with secondary antibodies (Li-COR Biosciences, Lincoln, NE, USA); an anti-mouse IgG conjugated with IRDye^®^800CW (catalog number 926-32210) (1:10000) or an anti-rabbit IgG conjugated with IRDye^®^680RT (catalog number 926-68071) (1:10000) at RT for 2 h. The immunoreactive bands were visualized using an Odyssey^®^ CLx Imaging System (LI-COR Biosciences, Lincoln, NE, USA). The bands were analyzed using the ImageJ software (developed at the National Institutes of Health, USA, http://rsb.info.nih.gov/ij).

### 4.8. Immunofluorescence Study

Immunofluorescence was performed to visualize NF-κB nuclear localization and Akt phosphorylation upon LPS stimulation. RAW 264.7 cells grown on glass cover slips were treated with AL extract for 3 h. Cells were then stimulated with 1 μg/mL of LPS for 30 min (for p-Akt staining) (antibody catalog number 4060, Cell Signaling Technology, Boston, MA, USA) and 24 h (for NF-κB staining) (antibody catalog number 8242, Cell Signaling Technology, Boston, MA, USA). After treatment, cells were fixed with 4% paraformaldehyde (Sigma-Aldrich, Saint Louis, MO, USA) dissolved in PBS for 15 min at RT. Cells were then washed three times with PBS for 5 min, each time. Next, cells were permeabilized with 0.3% Triton X-100 in PBS for 5 min. The cells were washed three times with PBS then blocked with 1% bovine serum BSA in PBS for 1 h. Cells were incubated with appropriate primary antibodies (Cell Signaling Technology, Boston, MA, USA) overnight at 4 °C. Primary antibodies included a rabbit anti-NF-κB antibody (1:400) and a phosphospecific rabbit anti-Akt (Ser473) antibody (1:400). After washing three times, cells were incubated with appropriate secondary antibodies at a 1:500 dilution of Alexa488-conjugated anti-rabbit IgG or Alexa594-conjugated goat anti-rabbit IgG (Thermo Fisher Scientific, Waltham (HQ), MA, USA) plus 10 μg/mL of Hoechst 33342 (Sigma-Aldrich, Saint Louis, MO, USA) or 1 μg/mL of DAPI (Sigma-Aldrich, Saint Louis, MO, USA) (for nuclear staining) for 2 h, in the dark at RT. Cells were washed three times with PBS (5 min each) and one time with distilled water (5 min). Finally, sample cover slips were mounted with Fluoromount-G (SouthernBiotech, Bermingham, AL, USA). The observations were performed on a fluorescence microscope, Axio Vert.A1 (Carl Zeiss, Oberkochen, Germany), with 100× magnification, and micrographs were captured with the Zen 2.6 (blue edition) Software for the Zeiss Axiocam 506 color microscope camera. 

### 4.9. Statistical Analysis

All data from the experiments were expressed as mean ± SD and then analyzed by one-way analysis of variance (ANOVA) with Tukey’s post hoc multiple comparisons on RAW data reads using SPSS software (SPSS Inc., Chicago, IL, USA). In all analyses, a *p*-value (*p* < 0.05) was considered as statistically significant.

## Figures and Tables

**Figure 1 ijms-21-01355-f001:**
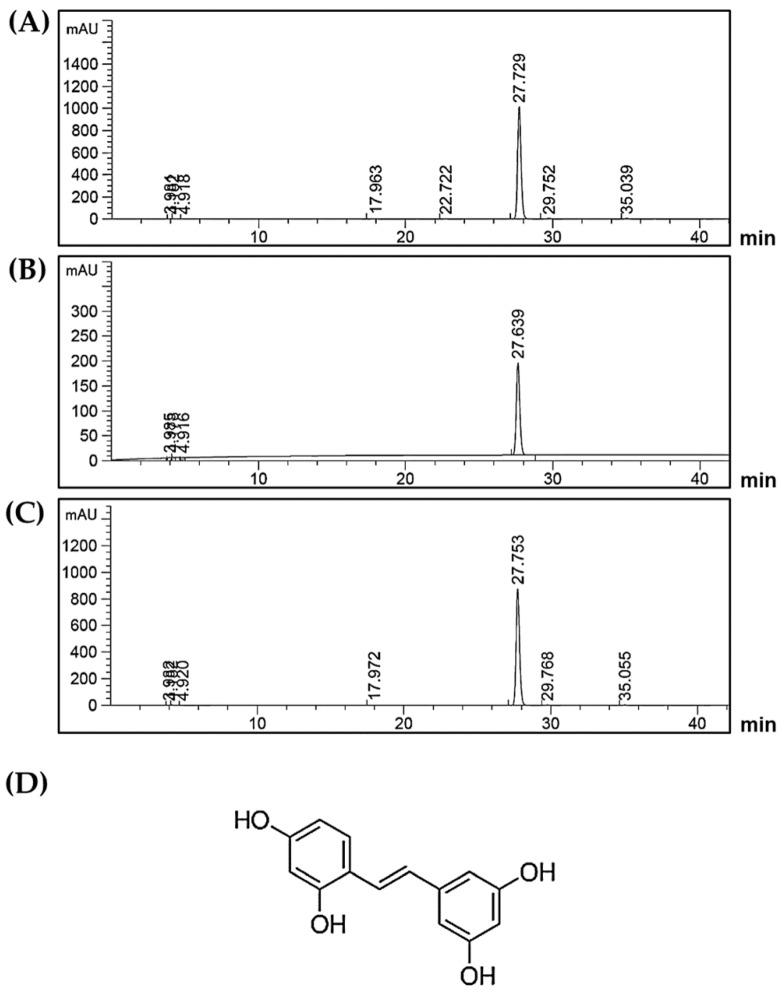
(**A**) HPLC chromatogram of *Artocarpus lakoocha* Roxb. (AL) ethanolic extract; (**B**) HPLC chromatogram of oxyresveratrol standard; (**C**) HPLC chromatogram of the mixture of AL ethanolic extract and oxyresveratrol standard; the retention time at 27 min indicates the existence of oxyresveratrol; (**D**) The chemical structure of oxyresveratrol.

**Figure 2 ijms-21-01355-f002:**
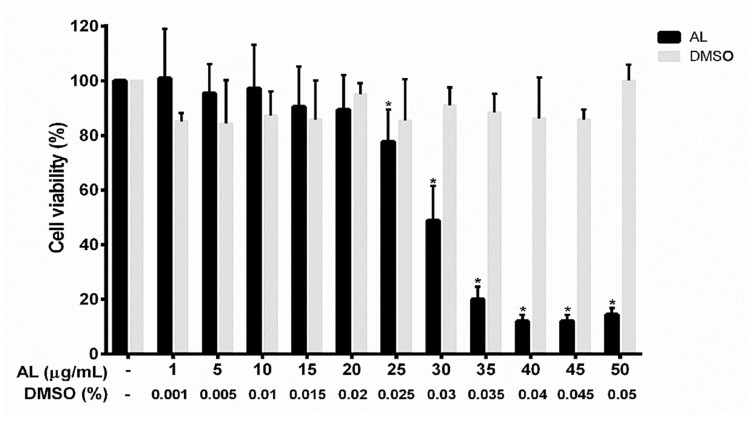
Effects of AL extract on cell viability in RAW 267.4 cells. The bars indicate percent cell viability of cells treated with AL extract with the indicated concentrations (ranging from 0 to 50 μg/mL) for 48 h, and subjected to 3-(4,5-dimethylthiazol-2-yl)-2,5-diphenyltetrazolium bromide (MTT) assay. Data are representatives of three replicates and shown as mean ± SD; * *p* < 0.05 (compared to the untreated group).

**Figure 3 ijms-21-01355-f003:**
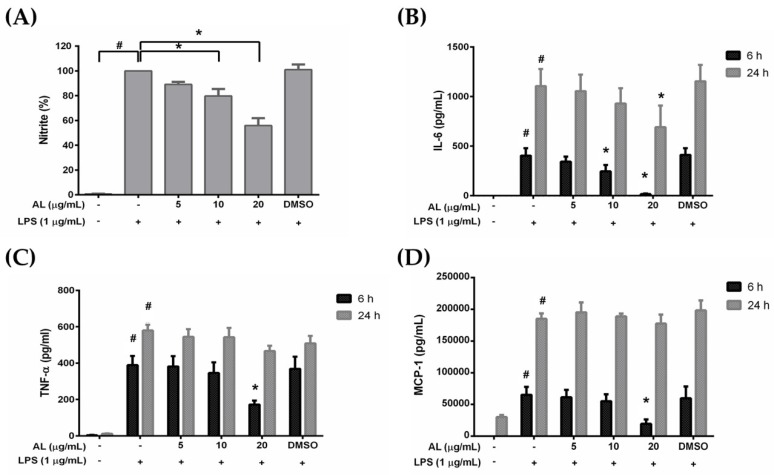
(**A**) The inhibitory effect of AL extract on lipopolysaccharide (LPS)-induced nitric oxide (NO) production in the supernatants of RAW 264.7 cells treated with different concentrations of AL extract (5, 10, and 20 μg/mL) for 24 h; (**B**) ELISA for the secreted level of IL-6 in the supernatants of cells treated with different concentrations of AL extract with the presence of LPS for 6 and 24 h; (**C**) ELISA for the secreted level of TNF-α in the supernatants of cells treated with different concentrations of AL extract with the presence of LPS for 6 and 24 h; (**D**) ELISA for the secreted level of MCP-1 in the supernatants of cells treated with different concentrations of AL extract with the presence of LPS for 6 and 24 h. Data are representatives of three replicates and shown as mean ± SD; ^#^
*p* < 0.05 (compared to the untreated control) or * *p* < 0.05 (compared to the LPS-treated group).

**Figure 4 ijms-21-01355-f004:**
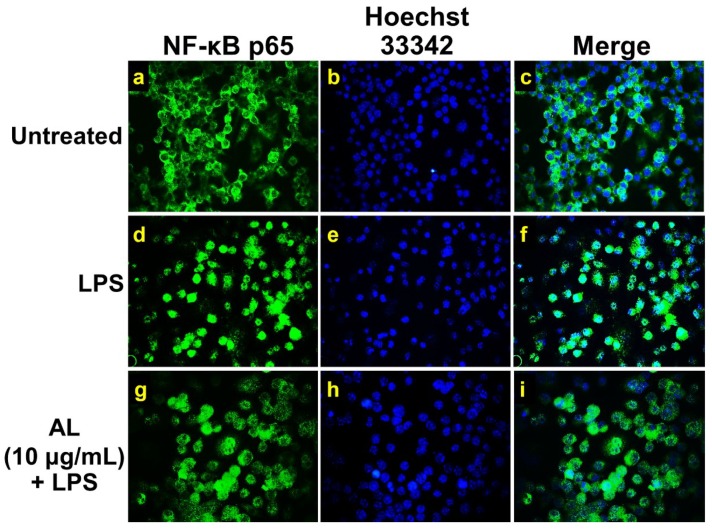
The inhibitory effects of AL extract on nuclear localization of nuclear factor-kappa B (NF-κB) in RAW 267.4 cells. Representative images from immunofluorescence study showing NF-κB staining (green) in untreated cells (**a**–**c**), LPS-stimulated cells (**d**–**f**), and AL-treated cells stimulated with LPS (**g**–**i**). Nuclei (blue) were stained with Hoechst 33342. The micrographs were captured at 100× magnification, and data are representatives of three replicates.

**Figure 5 ijms-21-01355-f005:**
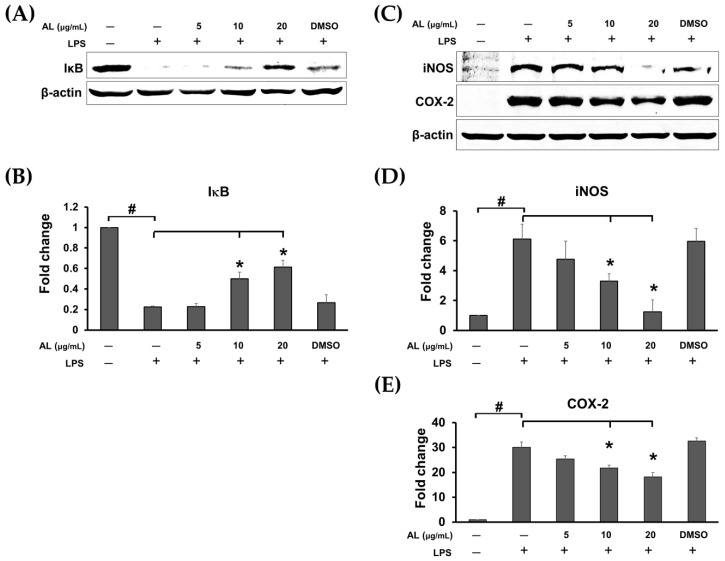
(**A**) Western blot analysis for the inhibitory effects of AL extract on the degradation of IκB; (**B**) quantitative analysis for IκB; (**C**) Western blot analysis for the inhibitory effects of AL extract on the protein expression level of iNOS and COX-2; (**D**) quantitative analysis for iNOS expression level; (**E**) quantitative analysis for COX-2 expression level. Beta actin (β-actin) was detected and used as an internal control. Data are representatives of three replicates and shown as mean ± SD; # *p* < 0.05 (compared to the untreated control) or * *p* < 0.05 (compared to the LPS-treated group).

**Figure 6 ijms-21-01355-f006:**
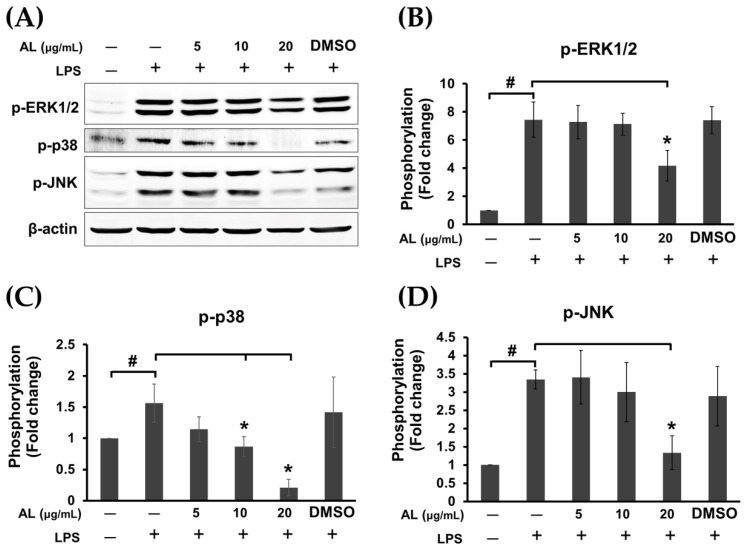
(**A**) Western blot analysis for the inhibitory effects of AL extract on the phosphorylation of mitogen-activated protein kinases (MAPKs), including ERK1/2, p-38, and JNK, in LPS-induced cells treated with AL extract at various concentrations for 24 h; degradation of IκB; (**B**) quantitative analysis for the phosphorylation of ERK1/2; (**C**) quantitative analysis for the phosphorylation of *p*-38; (**D**) quantitative analysis for the phosphorylation of JNK. Beta actin (β-actin) was detected and used as an internal control. Data are representatives of three replicates and shown as mean ± SD; ^#^
*p* < 0.05 (compared to the untreated control) or * *p* < 0.05 (compared to the LPS-treated group).

**Figure 7 ijms-21-01355-f007:**
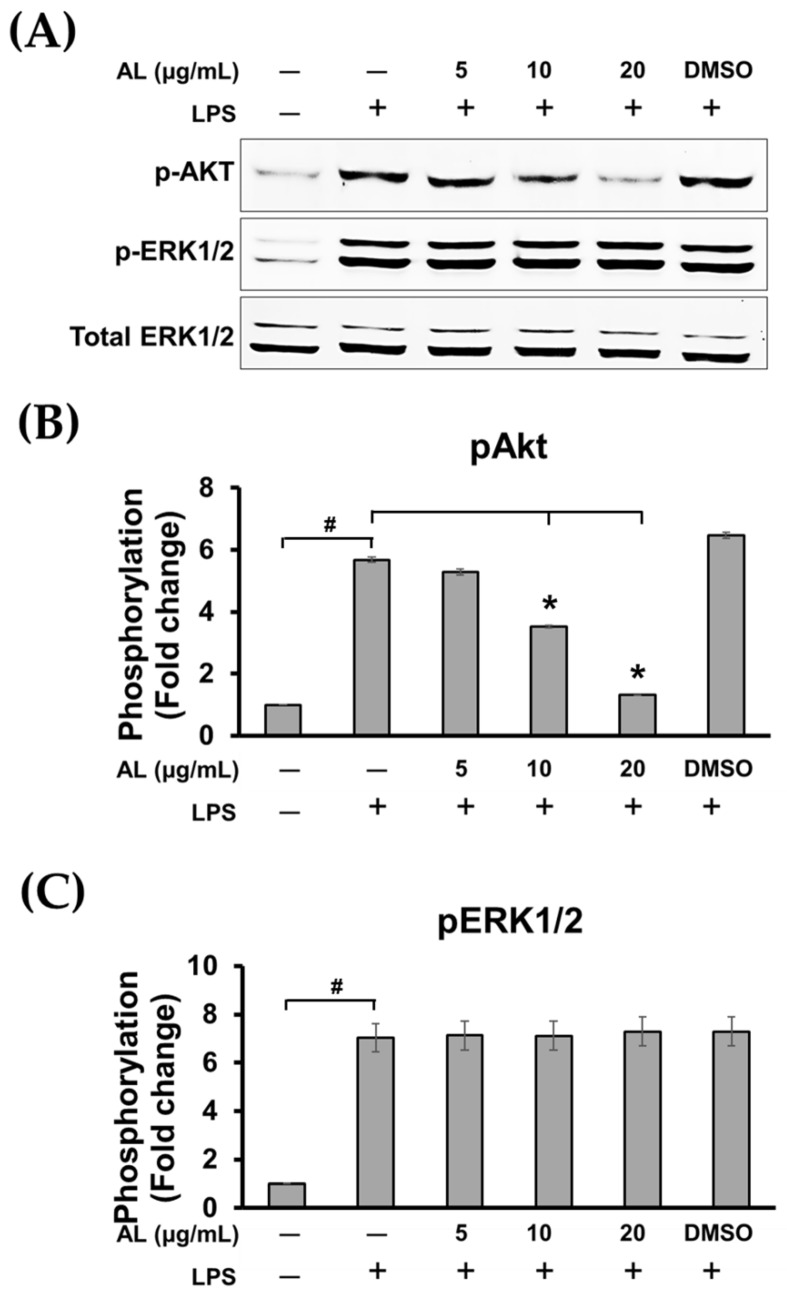
(**A**) Western blot analysis for the inhibitory effects of AL extract on the phosphorylation of Akt and ERK1/2 after cells were incubated with various concentrations of AL extract (5, 10, and 20 μg/mL) for 3 h before stimulation with LPS, total ERK1/2 was detected and used as an internal control; (**B**) quantitative analysis for Akt phosphorylation at Ser473; (**C**) quantitative analysis for ERK1/2 phosphorylation. Data are representatives of three replicates and shown as mean ± SD; ^#^
*p* < 0.05 (compared to the untreated control) or * *p* < 0.05 (compared to the LPS-treated group).

**Figure 8 ijms-21-01355-f008:**
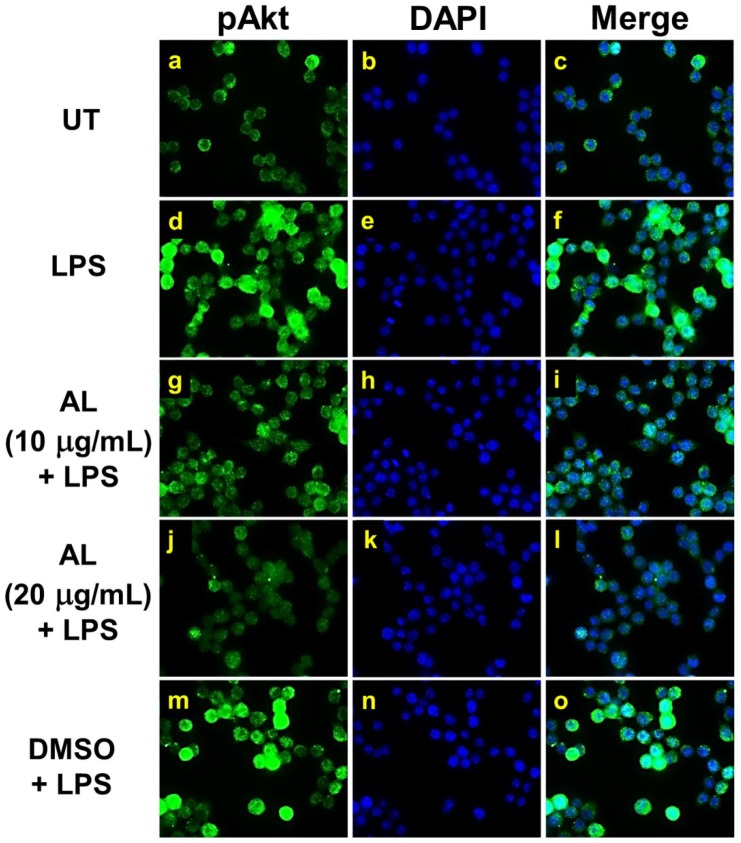
The inhibitory effects of AL extract on Akt phosphorylation in response to LPS induction in RAW 264.7 cells. Representative images from immunofluorescence study showing Akt phosphorylation at Ser473 (green) of untreated cells (**a**–**c**), LPS stimulated cells (**d**–**f**), LPS-stimulated cells with the presence of AL extract at 10 µg/mL (**g**–**i**) or 20 µg/mL (**j**-**l**), or DMSO (**m**–**o**). Nuclei (blue) were stained with DAPI. The micrographs were captured at 100× magnification, and data are representatives of three replicates.

**Figure 9 ijms-21-01355-f009:**
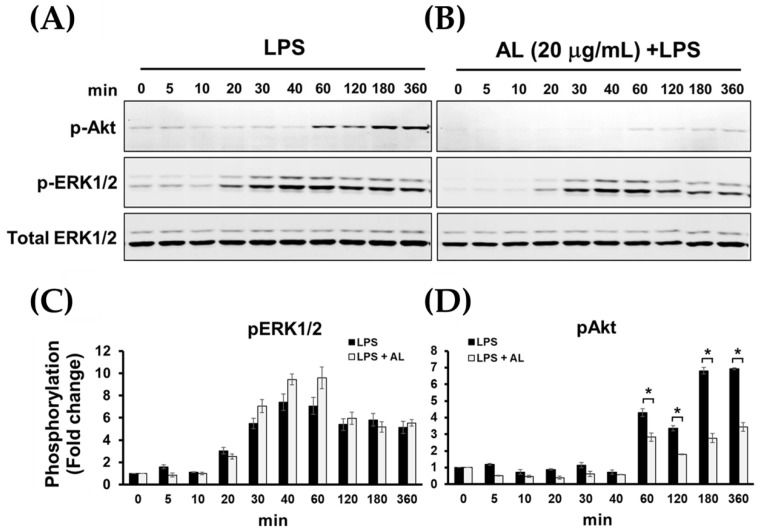
(**A**) Western blot analysis for the phosphorylation of Akt and ERK1/2 of RAW 264.7 cells induced with LPS for various time points (0–360 min); (**B**) the inhibitory effects of AL extract on the phosphorylation of Akt and ERK1/2 after LPS stimulation for various time points, total ERK1/2 was detected and used as an internal control; (**C**) quantitative analysis for ERK1/2 phosphorylation; (**D**) quantitative analysis for Akt phosphorylation at Ser473. Data are representatives of three replicates and shown as mean ± SD; * *p* < 0.05.

**Figure 10 ijms-21-01355-f010:**
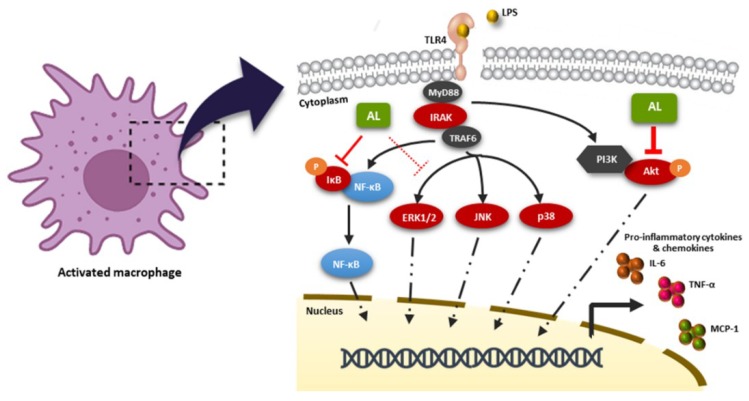
Schematic picture proposing that the AL extract regulates the LPS-stimulated inflammatory response. AL extract drastically inhibits the activation of the Akt and NF-κB signal transduction pathways, while moderately suppressing the MAPK signaling, resulting in the downregulation of inflammatory cytokine and chemokine production in response to LPS induction.

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
