# Peer review of "Artocarpus lakoocha* Extract Inhibits LPS-Induced Inflammatory Response in RAW 264.7 Macrophage Cells"

_ijms, 2020, doi:10.3390/ijms21041355_

Round 1

Reviewer 1 Report

It's a well written paper. The introduction is thorough, and goes into the inflammatory cascade in a comprehensive way. The research design is ace=ptable.
The manuscript didn’t display well, but I assume that it’s a technical issue. Please improve formatting. Otherwise, good article.

Author Response

Major points:

1. It's a well written paper. The introduction is thorough, and goes into the inflammatory cascade in a comprehensive way. The research design is acceptable.

Author:

We appreciate the comments from the reviewer. Thank you very much.

2. The manuscript didnt display well, but I assume that its a technical issue. Please improve formatting. Otherwise, good article.

Author:

We have checked the original file of our manuscript (both word and PDF files), and it is displayed find. I agree with the reviewer that it may be a technical issue during uploading the manuscript. When we re-submit the revised version of our manuscript, we will carefully perform the double check to ensure that the manuscript is displayed correctly.

Reviewer 2 Report

Authors presented evidence for the anti-inflammatory effects of AL extract in RAW264.7 cells.

The manuscript needs editing for formatting errors.

Line 60: activation by LPS can stimulate

For figure 5, DMSO control treatment is missing for all the data.

NO production assay is described in the methods but the data is not presented.

It will be interesting to see if oxyresveratrol demonstrates similar effects in macrophages when used as a purified chemical. This will also shed light on whether the effects demonstrated by the extract are merely due to oxyresvetatrol of any possible synergistic interactions in the extract. Authors should comment on the effects of the AL extract in comparison to purified chemicals isolated from the extract.

Author Response

Major points:

1. The manuscript needs editing for formatting errors.

Author:

We have checked the original file of our manuscript (both word and PDF files), and it is displayed find. I believe that it may be a technical issue during uploading the manuscript for our first submission.

However, when we re-submit the revised version of our manuscript, we will carefully double check to ensure that the manuscript is in the correct format.

2.Line 60: activation by LPS can stimulate

Author:

We thank the reviewer very much for pointing this error. We have changed the sentence as suggested, and all changed areas have been highlighted in yellow and can be seen byTrack Changes”.

3. For figure 5, DMSO control treatment is missing for all the data.

Author:

We have added DMSO control into the figure as mentioned and replaced the old picture with the new one. Thank you very much.

4. NO production assay is described in the methods but the data is not presented.

Author:

Since nitric oxide (NO) can actually be measured by the amount of nitrite because nitrite is used as an indicator of NO production, we presented the data as the level of nitrite.  The results for NO detection was originally displayed as the percentage of nitrite which was presented in figure 3A.

5. It will be interesting to see if oxyresveratrol demonstrates similar effects in macrophages when used as a purified chemical. This will also shed light on whether the effects demonstrated by the extract are merely due to oxyresveratrol of any possible synergistic interactions in the extract. Authors should comment on the effects of the AL extract in comparison to purified chemicals isolated from the extract.

Author:

We totally agree with the reviewer on this point. We believe that adding more complex evidence to clearly verify whether the observed results are caused mainly by oxyresveratrol, would be valuable to increase the scientific and clinical significance, especially when we develop potential agent to be used in humans in the future. We have a next-step plan for comparative study of AL extract and the purified oxyresveratrol (both from the plant and from the commercially available sources) in several different macrophage cells and microglial cells including THP-1, BV-2, and HMC3 cells.

However, the current study focuses mainly on exploring whether AL extract which contains more than 70% of oxyresveratrol possesses the anti-inflammatory properties. We will definitely continue our work as constructively suggested by the reviewer.

Round 2

Reviewer 1 Report

Changes acceptable. A good article!

Reviewer 2 Report

Accept in the present form.